# Prognostic Markers of Myelodysplastic Syndromes

**DOI:** 10.3390/medicina56080376

**Published:** 2020-07-27

**Authors:** Yuliya Andreevna Veryaskina, Sergei Evgenievich Titov, Igor Borisovich Kovynev, Tatiana Ivanovna Pospelova, Igor Fyodorovich Zhimulev

**Affiliations:** 1Laboratory of Gene Engineering, Institute of Cytology and Genetics, SB RAS, 630090 Novosibirsk, Russia; 2Department of the Structure and Function of Chromosomes, Laboratory of Molecular Genetics, Institute of Molecular and Cellular Biology, SB RAS, 630090 Novosibirsk, Russia; titovse78@gmail.com (S.E.T.); zhimulev@mcb.nsc.ru (I.F.Z.); 3Vector-Best, 630117 Novosibirsk, Russia; 4Department of Therapy, Hematology and Transfusiology, Novosibirsk State Medical University, 630091 Novosibirsk, Russia; kovin_gem@mail.ru (I.B.K.); depart04@mail.ru (T.I.P.)

**Keywords:** myelodysplastic syndrome, IPSS-R, karyotype, acute myeloid leukemia, microRNA

## Abstract

Myelodysplastic syndrome (MDS) is a clonal disease characterized by multilineage dysplasia, peripheral blood cytopenias, and a high risk of transformation to acute myeloid leukemia. In theory, from clonal hematopoiesis of indeterminate potential to hematologic malignancies, there is a complex interplay between genetic and epigenetic factors, including miRNA. In practice, karyotype analysis assigns patients to different prognostic groups, and mutations are often associated with a particular disease phenotype. Among myeloproliferative disorders, secondary MDS is a group of special entities with a typical spectrum of genetic mutations and cytogenetic rearrangements resembling those in de novo MDS. This overview analyzes the present prognostic systems of MDS and the most recent efforts in the search for genetic and epigenetic markers for the diagnosis and prognosis of MDS.

## 1. Introduction

Myelodysplastic syndromes (MDS) are a group of clonal diseases of hematopoietic stem cells and are characterized by multilineage dysplasia in immature myeloid cells, ineffective hematopoiesis, peripheral blood cytopenias and a high risk of transformation to acute myeloid leukemia (AML) [1]. A retrospective analysis of 36,558 MDS cases revealed that the frequency of AML secondary to MDS was 3.7% in patients aged 40 years and under, and 2.5% in patients aged 40 years and over [2]. Although no epidemiological data on MDS in Russia have been gathered yet, the Surveillance, Epidemiology and End Results (SEER) data suggests that the majority of MDS patients in the USA with an onset in 2012–2016 were above 70 years of age [3].

In most cases, primary MDS is an aging-associated disease, while secondary MDS is quite a common occurrence in the youth [4]. Secondary MDS is associated with symptomatic peripheral blood (PB) cytopenias and abnormal bone marrow (BM) cell morphology resembling those observed in primary MDS. “Secondary MDS” has long been a name for the MDS that cancer patients develop following treatment with cytostatic agents (t-MDS) [5]. However, in practice, secondary MDS may well be developed by people with untreated autoimmune diseases and solid-type neoplasms [6].

At present, the best practices of diagnosing MDS include a comprehensive assessment of morphological data on biopsied BM samples, cytogenetic data, BM cell immunophenotyping data and PB test results. It is equally important to assess quantitative and qualitative changes. Diagnosing MDS is challenging; for example, at early stages, it is quite difficult to differentiate minor morphological changes associated with dysplasia from other BM deficiencies [7]. As any MDS diagnosis relies on morphological evidence, opinions may be inconsistent [8]. The proliferation of blast cell and changes to the BM cell composition in primary MDS followed by the emergence of a pathological cell clone take place due to genetic and epigenetic factors [9].

Significant advances in treatment for MDS have now been made. However, choosing the optimal moment and integrating allo-HSCT into the therapeutic algorithm remains a challenge in many cases, and also, the response to treatment is adversely affected by hypomethylating agent resistance [10]. Analysis of cytogenetic and morphological data allows MDS patients to be assigned to different groups according to the outcome predictions [11]. However, the course of this disease may be variable even between patients of the same prognostic subgroup. In practice, survival is not always high for favorable-prognosis patients, while the poor-prognosis group may reveal individuals with an indolent course. To improve diagnostic and prognostic accuracy in assessing MDS, the search for additional molecular-genetic markers is required.

The aim of this work is to give an overview of the current prognostic systems of MDS and the most recent efforts in the search for genetic and epigenetic markers for the diagnosis and prognosis of MDS.

## 2. MDS Classification

MDS is a group of hematopoietic disorders involving myeloid lineage cells. One of the former names for MDS is “preleukemia” [12]. MDS is clonal BM neoplasms and includes a heterogeneous group of myeloid neoplasms characterized by one or more PB cytopenias, and morphological dysplasia in one or more hematopoietic lineages [1]. The estimated degree of cytopenia is another important prognostic factor in addition to the presence itself of this pathology [13]. It has been proven that the degree of anemia and thrombocytopenia is associated with a poor clinical prognosis [14,15].

### 2.1. MDS-Like Idiopathic Changes of Indeterminate Potential

To be diagnosed with MDS, one has to have dysplastic cells in excess of 10% [16,17,18]. In cases with low-grade dysplasia or no dysplasia, other potential causes of cytopenia should be excluded [19]. Cytopenic patients without dysplasia are in the group named “idiopathic cytopenia of undetermined significance” (ICUS). Some of them may have cytogenetic abnormalities, which co-occur with one or more somatic mutations in up to 40% of cases–these patients are in the subgroup named “clonal cytopenia of undetermined significance” (CCUS). About 25% of patients with ICUS may eventually develop MDS or AML, but the risk is much higher if a clonal mutation is present [20]. Dysplastic patients with low-grade cytopenia are in the subgroup named “idiopathic dysplasia of uncertain significance” (IDUS). Additionally, there are cases of somatic mutations in elderly patients’ hematopoietic cells without hematologic abnormalities, the so-called age-related clonal hematopoiesis of indeterminate potential (CHIP) [21]. In most cases, this condition is noted for the presence of single mutations in the following genes: *DNMT3A, TET2, ASXL1, SF3B1, SRSF2, TP53, CBL,* and *U2AF1*. Other observations in CHIP are cytogenetic abnormalities, such as monosomy 7 and trisomy 8 (Table 1) [22]. Although some patients with CHIP do develop MDS later in their life, the pathogenesis of this condition is not yet fully understood, and so the sole presence of somatic mutations is not considered diagnostic of MDS in the cases as these.

### 2.2. The FAB Classification

The first attempt to classify MDS cases in a standardized way, indicating the risk of transformation to AML, came to the scene with the advent of the French-American-British (FAB) classification (1982), which divided MDS into five subtypes based on PB and BM morphology: refractory anemia (RA), refractory anemia with ringed sideroblasts (RARS), refractory anemia with excess of blasts (RAEB), refractory anemia with excess of blasts in transformation (RAEBt) and chronic myelomonocytic leukemia (CMML) (Table 2) [23].

The term “refractory” is used here to describe the following situation: treatment of this anemia with iron preparations, vitamin B12 and other nutrients is ineffective, myeloblasts make up less than 5% of bone marrow cells and abnormalities are largely observed in red cell precursors. RARS is characterized by myeloblasts making up less than 5% of bone marrow cells, but no less than 15% of red cell precursors appear as specific abnormal cells called “ringed sideroblasts”. RAEB is characterized by myeloblasts making up 5–20% of bone marrow cells. RAEBt is characterized by 21–30% myeloblasts. According to this classification system, MDS is diagnosed if there is evidence of dysplasia and/or myeloblasts make up 5–30% of all bone marrow cells. Childhood MDS and secondary MDS have no clear criteria in the FAB classification. Naturally, as time went by, the need for the MDS classification system to be improved was becoming more and more evident.

### 2.3. The WHO Classification

Today, the system adopted by the WHO has become the recognized standard for the MDS classification, with some of FAB elements retained and MDS categories expanded [24]. The WHO classification of MDS was developed in 1999 and was updated is 2001, 2008 and 2016 (Table 2). The WHO classification is superior to the FAB classification and is included in the World Health Organization classification-based Prognostic Scoring System (WPSS) [13]. The WHO system features the following new diagnostic subtypes: MDS with a deletion involving the long arm of chromosome 5 and differentiating between single-lineage and multilineage dysplasia within RA. RAEBt was excluded, for its treatment options resemble those for AML. CMML was excluded, for its clinical manifestations resemble those of myeloproliferative disorders. According to the WHO classification, patients with ≥20% bone marrow blasts are AML cases, while the former cut-off was ≥30%. Therefore, the choice of treatment strategy for patients with bone marrow blasts in the range between 20% and 30% should be personalized in the most precise manner. Unclassifiable MDS and childhood MDS imply <10% of dysplastic cells of one or more myelopoietic lineages in the presence of a cytogenetic abnormality that is considered presumptive evidence of MDS. Additionally, the WHO classification in its current version singles out AML with myelodysplasia-related changes, which is for patients who developed AML after MDS, AML with multilineage dysplasia and AML with MDS-related cytogenetic abnormalities. AML secondary to MDS, too, is noted for cytogenetic abnormalities and mutations, many of which are identical to those in de novo MDS (Table 1) [25,26,27,28]. In independent validations, the WHO revisions were shown to provide more-homogeneous subgroups of patients and greater prognostic power compared with the FAB system, although controversies remain [29]. Navarro et al. analyzed the prognostic impact of the WHO and FAB morphologic classifications in a series of patients with primary MDS. They show that 17.7% patients with MDS, according to FAB criteria, no longer presented MDS after reclassification according to the WHO proposals [30]. In the other study, 103 cases of MDS previously classified by FAB were reclassified according to the WHO proposal. There was a significant interobserver agreement and discrepancies among observers nearly always related to the identification and enumeration of dyspoiesis in neutrophils and megakaryocytes. The present study suggests that in patients with less than 5% blasts, an important negative variable for response to treatment as well as for survival is nonerythroid multilineage dysplasia [31].

## 3. Genetic Changes in MDS

An important component of MDS in the definition given it by the WHO is the clonal nature of myelodysplastic hematopoiesis [24]. Additional mutations lead to the formation of subclones of hematopoietic cells with subsequent differentiation abnormalities, an increase in blasts cells, and, consequently, AML. The number of mutations varies depending on the disease subtype and increases as the disease progresses [32]. The timing of mutations is not fixed, it varies from one individual to the next. Thus, the same mutation can be an early event in some patients and subclonal in the others. However, mutations of *SF3B1, U2AF1*, and *TP53* are more likely to be dominant, and those of *ASXL1, CBL*, and *KRAS* are secondary [33].

A large number of works mentioning genome instability in MDS have been published (Table 1) [34,35]. Thus, analysis of mutations in MDS showed that 89.5% of cases have at least one mutation each, of them 67.9% have a normal karyotype. Mutations to the *TET2, SF3B1, ASXL1, SRSF2, DNMT3A* and *RUNX1* were most frequent. Relatively rare mutations, with a frequency of below 10%, were observed in the *U2AF1, ZRSR2, STAG2, TP53, EZH2, CBL, JAK2, BCOR, IDH2, NRAS, MPL, NF1, ATM, IDH1, KRAS, PHF6, BRCC3, ETV6* and *LAMB4* genes [36].

### 3.1. Mutations to Components of the Splicing Machinery in MDS

Somatic mutations to genes involved in splicing are observed in half of MDS cases. The main splicing-related mutations revealed in MDS occur in *SF3B1, SRSF2, U2AF35* and *ZRSR2* [37]. Mutations to these genes cause mistakes in RNA splicing, leading to the assembly of intron-containing unspliced RNA. *DDX41* and *LUC7L2* are involved in splicing regulation, and somatic mutation in these genes have been registered in MDS too, albeit not so frequently [38]. Different mutations in the genes encoding components of the splicing machinery are associated with different MDS phenotypes and different clinical outcomes. Mutations in *SF3B1* occur largely in refractory anemia with ringed sideroblasts, and the overwhelming majority of such cases with *SF3B1* mutants have a favorable clinical prognosis with a low risk of transformation to AML [39]. A meta-analysis on nine studies involving a total of 2259 MDS cases showed that *SF3B1* mutations have no effect on overall survival in these patients [40]. Mutations and deletions in *PRPF8* are observed in less than 5% of MDS cases and correlate with ringed sideroblasts but are associated with a more aggressive course of the disease [41]. Mutations in *SRSF2* occur in about 10% of MDS cases and are observed together with neutropenia and pronounced thrombocytopenia, largely in patients with multilineage dysplasia and/or excess blasts and are prognostic of a high risk of AML and poor overall survival [42]. Mutations in *U2AF1* do not correlate with MDS subtypes and are prognostic of a high risk of AML and shorter overall survival [43].

### 3.2. Mutations in Epigenetic Regulatory Systems in MDS

Methylation is one of the mechanisms involved in epigenetic regulation of transcription in solid-type neoplasms and hematopoietic tumors [44]. An important role in DNA methylation lies with the *DNMT* (*DNMT3A, DNMT3B* and *DNMT1*), *TET* (*TET2, TET2* and *TET3*) and IDH (*IDH1/IDH2*) gene families. One of the first mutations to be discovered in MDS was in TET2, this mutation being detected in ~25% of MDS patients [45]. Mutations in *IDH1/2* are observed in less than 10% of MDS patients and in 20% of AML patients, are associated with a poor clinical outcome, and seldom do they occur at once [46]. In AML, the concurrence of *NPM1* and *IDH1* or *IDH2* mutations is associated with a favorable clinical outcome; however, in MDS, this is associated with poor overall survival [47]. A meta-analysis involving 2236 MDS cases showed that patients with *DNMT3A* mutations had a much poorer prognosis than had patients without these mutations [48].

Histone modification plays an important role in chromatin remodeling and gene expression. *EZH2* encodes PRC2’s catalytic subunit, which mediates chromatin compaction and catalyzes histone H3K27 methylation. It has been found that somatic mutations in *EZH2* occur in 6% of MDS cases and are an independent factor of poor prognosis [49]. *EZH2* is located in 7q36.1, which is normally deleted in MDS, and that is how this gene has lost its function: either due to a mutation or due to the chromosomal deletion. Mutations in *ASXL1*, which is presumably involved in chromatin modification, are observed in ~10% of MDS cases, impair the myeloid differentiation of hematopoietic cells and reduce median survival [50].

### 3.3. Mutations to TP53 in MDS

*TP53* is a tumor suppressor gene and a transcription factor regulating apoptotic pathways, cell cycle arrests and DNA repair. Mutations to this gene are generally associated with high-risk MDS, including MDS with excess blasts and treatment-induced myeloid neoplasms [51]. The concurrence of a *TP53* mutant and del(5q) is observed in 15% of MDS cases and contributes to resistance to treatment [52].

### 3.4. Inherited Mutations in MDS

Germline mutations can be a stronger influence on the development of myeloid malignant neoplasms than were believed. Traditionally, familial MDS are considered rare, especially in adulthood; however, the growing accessibility of genetic testing incorporated into clinical practice has enabled identification of a large number of mutations associated with a predisposition to MDS, including those in *RUNX1, ANKRD26, DDX41, ETV6, GATA2* and *SRP72* [53,54]. In 2012, 27 families with familial MDS/AML were screened for mutations in *RUNX1, CEBPA, TERC, TERT, GATA2, TET2* and *NPM1*, and mutations were found in ten of them [55].

## 4. Secondary MDS

Therapy-related myeloid neoplasms, in particular, the so-called secondary MDS, most commonly appear as complications following the treatment of primary tumors and autoimmune diseases with cytostatic agents, for example, Anthracyclines [56]. As de novo MDS, secondary t-MDS is noted for a number of normally detectable mutations and cytogenetic abnormalities, with genetic abnormalities being detected in half of de novo MDS cases and in nearly 90% of secondary MDS cases (Table 1) [35,57]. The most frequent abnormalities in secondary MDS are −5/del(5q), −7/del(7q), trisomy 8 and monosomy 18. Compared to de novo MDS cases, patients with secondary MDS exhibit a higher frequency of high-risk genetic abnormalities associated with poor survival. Additionally, del(5q) is a frequent observation in radiation therapy-related t-MDS, while monosomy 7 is more common following treatment with alkylators [35]. There are two types of t-MDS/AML recognized by the WHO classification depending on the causative therapeutic exposure: an alkylating agent/radiation-related type and a topoisomerase II inhibitor-related type. Both differ in their cytogenetic abnormalities. The first type is associated with involving chromosomes 5 (−5/del(5q)) and 7 (−7/del(7q)), while the latter is associated with balanced translocations involving chromosome bands 11q23 or 21q22 [58]. Patients with secondary MDS may typically have cytogenetic rearrangements or somatic mutations. Thus, mutations to *TP53* and *IDH1* were much more frequent in secondary MDS than in de novo MDS [59]. Mutations to the *SF3B1, U2AF1, SRSF2* and *ZRSR2* genes regulating RNA splicing were detected in ~50% of patients with secondary MDS with ringed sideroblasts and in nearly 90% of patients with de novo MDS [60]. Westman and the co-workers described an increased frequency of mutations to *IDH1* and *IDH2* in t-MDS patients with the translocation der(1;7)(q10;p10) [61]. In another work, the translocation t(6;15)(q12;q15) was detected in a patient who had developed t-MDS after having normal-karyotype AML. The patient had a good response to treatment, suggesting that this rearrangement may have a prognostic value [62]. Stating secondary MDS has long implied the presence of morphological changes induced in bone marrow cells by past treatment. However, later cases of secondary MDS were reported in patients with malignant lymphomas before the onset of treatment, and nonleukemic patients with non-Hodgkin lymphomas display cytomorphological evidence of secondary dyserythropoiesis [6,63].

## 5. Prognostic Systems for MDS

Prognostic systems are useful instruments for assessing the risk of leukemia, patients’ lifespan and for personalizing therapy in the most precise manner. There are three MDS assessment and prognostic systems currently most in use: International Prognostic Scoring System (IPSS), Revised International Prognostic Scoring System (IPSS-R) and WHO-classification based prognostic scoring system (WPSS) [11,13,64]. However, their results are not always consistent; additionally, they may depend on the particular human population (for example, IPSS-R worked better in China) [65]. Noteworthy, prognostic models handling MDS target de novo MDS rather than secondary MDS. Nevertheless, Zeidan and the co-workers showed that IPSS-R can make accurate predictions even for secondary MDS [66]. Nazha et al., created a new model that incorporates mutational data to improve the predictive capacity of the IPSS-R in treated MDS patients. Independent significant prognostic factors for survival included age, IPSS-R and mutations *EZH2, SF3B1, TP53*, which associated with worse overall survival (OS) [67].

### 5.1. The International Prognostic Scoring System (IPSS)

Greenberg and the co-workers developed the IPSS, which included such variables as percentage of bone marrow blast cells, cytogenetic abnormalities and number of peripheral blood cytopenias [64]. IPSS separates MDS patients into four different risk subgroups: low, intermediate-1, intermediate-2 and high. One of IPSS limitations is that it was developed for assessing prognosis once the diagnosis has been rather than for use at any time throughout disease progression. Although IPSS is not perfect, it gives accurate estimates of survival and of the risk of evolving AML in primary MDS. Karyotype has become one of the most important prognostic factors in MDS. Unlike AML, with a predominance of balanced abnormalities, MDS has mostly unbalanced rearrangements [24]. Some of the most frequent cytogenetic abnormalities in MDS are del(5q), −7/del(7q), +8, del(20q) and −Y [68]. Regarding prognosis, IPSS classified the karyotype into three risk groups: “good” for diploid, del(20q), del(5q) and −Y; “intermediate”; and “poor” for chromosome 7 anomalies and complex karyotypes [64]. However, later studies using larger databases on MDS patients showed that some of rare chromosome abnormalities that are normally classified as intermediate have different prognostic relevance [68].

### 5.2. The Revised International Prognostic Scoring System (IPSS-R)

The IPSS-R was published in 2012; it classifies patients into five subgroups for risk, including the “intermediate” group (Table 3). The new categories in the current version of the system include 5 rather than 3 cytogenetic prognostic subgroups and 5 rather than 4 main prognostic categories as in IPSS. IPSS-R classifies patients into 5 cytogenetic categories, which have prognostic relevance based on overall survival and the risk of transformation to AML from “very good” to “very poor” [11].

#### 5.2.1. “Very Good”

Loss of the Y chromosome is normally associated with age, but this can also happen due to hematologic malignancies. If loss of the Y chromosome is the only abnormality in an MDS case, the prognosis is very good [69,70]. Chromosome 11 is one of the highest gene density chromosomes in the human genome [71]. Wang and the co-workers point out that del(11q) correlates with ringed sideroblasts, bone marrow hypocellularity and less severe thrombocytopenia and thus may possibly be a favorable prognostic indicator [72].

#### 5.2.2. “Good”

Deletion of the long arm of chromosome 5, del(5q), is observed in ~20% of MDS patients; however, only 5% of cases are classified as del(5q) MDS, while most cases display excess BM blasts and/or other cytogenetic abnormalities worsening the prognosis. For MDS patients with del(5q) and only one additional abnormality, prognostic results are conflicting and in part depend on the type of additional abnormality [73]. Mutations to TP53 are observed in ~20% of MDS patients with del(5q), and most of them have a higher risk of evolving AML and poorer overall survival [74]. Jerez and the co-workers demonstrated that patients with deletions of the centromeric and telomeric extremes of 5q have a more aggressive disease phenotype [75]. Molecular studies demonstrated that allelic haplodeficiency for several genes located in the removable part of 5q (in particular, *CSNK1A1, RPS14, EGR1*, *miRNA-145* and *miRNA-146a*) underlies both hematologic phenotype and selective sensitivity to treatment [76,77]. It has been pointed out that del(5q) MDS is associated with excessive telomere length shortening as a driving force for clonal evolution and AML progression [78]. Although MDS transforms to AML often, it can also transform to other leukemias, including acute lymphocytic leukemia (ALL). The presence of del(5q) and mutations in *TP53* or *EZH2* are associated with the emergence of the ALL clone [79]. Deletion of the long arm of chromosome 20, too, is an occurrence in MDS and is detected at a frequency of 3–7%, often co-occurring with anemia and thrombocytopenia [80,81]. Patients with this genetic abnormality have a relatively favorable prognosis [82].

#### 5.2.3. “Intermediate”

Monosomy 7 and del(7q) occur at an identical frequency: in 10% of de novo MDS and 50% of therapy-related MDS cases, and the presence of either alone is associated with a poor prognosis [38]. However, there is a hypothesis stating that clinical features associated with sole del(7q) are different from those associated with a complete loss of chromosome 7 [83]. Trisomy 8 is a cytogenetic abnormality typical of 5–7% of MDS cases and is classified as the intermediate cytogenetic risk group. This abnormality is considered a secondary or late event in MDS transformation, with exact mechanisms underlying its contribution to oncogenesis remaining unclear [84]. The median survival for patients with trisomy 8 is between 11 and 25 months, while patients with ≥5% blasts in BM in combination with trisomy 8 have a relatively shorter survival and an increased risk of AML transformation [85]. In MDS, deletion 13q occurs at a frequency of about 2% [86]. It has been hypothesized that unclassifiable MDS with del(13q) is a benign bone marrow failure subset with a good response to immunosuppressive therapy and a ten-year survival rate of about 80% [87]. It should be noted that IPSS-R factors in some less frequent abnormalities occurring in less than 1% of MDS cases. Trisomy 11 and trisomy 13 are very rare pathologies in MDS, each occurring at a frequency of 0.5%. In MDS patients with trisomy 11, the median survival is only 14 months and 69% of affected people develop secondary AML in a median period of 5 months, but because this rearrangement occurs at a very low frequency, it is still assigned to the intermediate-risk group [88]. Translocation 11q23/MLL is another very rare occurrence in primary MDS, with the overall survival being about 26 months and the risk to develop acute leukemia being up to 40% within a one-year interval and up to 92% within a five-year period. Loss of one X chromosome in women is a rare abnormality in MDS and AML: it occurs in 0.2–0.3% of cases and in up to 1.5% of patients who combine these conditions with other abnormalities and is characterized by a median overall survival of ~6 months [89]. The presence of translocation 1q in three patients originally diagnosed with MDS, which later evolved to AML, has been reported [90]. Translocation der(1;7)(q10;p10) is a rare rearrangement and is assigned to the intermediate risk-group due to the lack of adequate clinical data. This translocation is associated with profound thrombocytopenia, which most probably reduces median overall survival to 26 months [91]. Translocation der(5;17)(p10;q10) is another rare observation in MDS, largely occurring within a complex karyotype [92]. All rare cytogenetic abnormalities are assigned by the IPSS-R to the intermediate risk-group. However, because very few patients are so affected, this prognostic estimate should be considered with caution.

#### 5.2.4. “Poor” and “Very Poor”

It should be noted that complex karyotype (CK) in MDS cases, which is characterized by the presence of at least three clonal abnormalities and is especially common in secondary MDS, normally refers to both quantitative and structural changes. Complex karyotype may be applied to about 10-15% of MDS patients and is associated with a very short median survival and a high risk of AML transformation [93]. Complex karyotypes in MDS can result from the gradual acquisition of genetic changes by separate cells in the course of clonal evolution. Cases with five and more karyotypic abnormalities were described as having “high complexity”; they occur much more commonly in the presence of TP53 mutations [94]. An important part of a cytogenetic analysis of MDS is to consider the so-called “monosomal karyotype” (MK), which is defined as two or more autosomal monosomies or one monosomy with at least one additional structural rearrangement. It has been reported that MK negatively affects overall survival in patients who have underwent allogeneic hematopoietic stem cell transplantation and in MDS cases with CK [95,96]. Nevertheless, MK is not considered by the IPSS-R as a separate prognostic factor.

### 5.3. The WHO Classification-Based Prognostic Scoring System (WPSS)

The WPSS is a “time-dependent” scoring system, which dynamically analyzes the situation as the disease progresses and a patient is going through treatment [14]. With the use of WHO categories, cytogenetics, and red blood cell dependency, WPSS is a system similar to IPSS-R in that both classify patients into five prognostic groups from very low risk to very high risk [13]. However, even WPSS has some important limitations: patients with secondary MDS are excluded, cytogenetic categories are not optimally delineated, neither age nor somatic status is considered. Comorbidities affect low-risk and high-risk patients with MDS differently. In low-risk patients with MDS, comorbidities affect prognosis by directly increasing the risk of death that is not associated with transformation to acute leukemia. Additionally, and conversely, in their high-risk peers, any non-severe comorbid condition is not so clinically significant as a poor prognosis for MDS itself. Thus, reliance on the factors that have relevance to MDS and comorbid conditions substantially enhances the prognostic potential of the scoring systems used, especially in regard of low-risk groups. The model of clonal evolution spanning from CHIP to a hematologic malignancy assumes a complex interaction between epigenetic changes, a dysfunctional BM microenvironment and a step-by-step acquisition of additional key mutations. A comprehensive approach to the diagnosis of MDS, with a thorough understanding of genetic and epigenetic mechanisms of tumor development, will allow treatment to be personalized in the most precise manner. In particular, some of the promising markers for the diagnosis and prognosis of MDS are miRNAs, small noncoding RNAs regulating hundreds of genes and participating in a large number of pathways of disease initiation, development and progression.

Gene mutations have not yet been included in the 2016 WHO classification and IPSS-R. Mutational analyses showed that mutations of *CBL, IDH2, ASXL1, DNMT3A*, and *TP53* were independently associated with shorter survival. Patients within each IPSS-R or 2016 WHO classification-defined risk group could be stratified into two risk subgroups based on the mutational status of these five genes; patients with these poor-risk mutations had an OS shorter than others in the same risk group, but similar to those with the next higher risk category [97].

## 6. MiRNA in the Prognosis of MDS

Every stage of hematopoiesis is regulated by genetic and epigenetic factors, including miRNAs [98]. MiRNAs are among the regulators of normal hematopoiesis, and it is not surprising that changes in their expression levels contribute to the development of hematologic neoplasms. MDS is regarded as a disease preceding leukemia and about 30% of patients with MDS eventually develop AML [24]. Analysis of literature data showed that miRNA expression profiles differ between early and advanced stages of MDS, suggesting the involvement of miRNAs in the pathogenesis of MDS and, consequently, in MDS-to-AML transformation [99]. The expression levels of miRNA-27a-3p, −150-5p, −199a-5p, −223-3p and miRNA-451a are reduced in the plasma of high-risk patients with MDS, while miRNA-196b-5p is enhanced in BM specimens from a higher-risk patient with MDS, suggesting that these miRNAs may be used as biomarkers predicting the risk of further transformation of MDS [100,101]. Elevated miRNA-661 and miRNA-125a are negatively correlated with survival in MDS [102,103]. Patients with low expression levels of miRNA-194-5p are noted for a substantial reduction in overall survival as well [104]. Profiling of the expression of miRNA circulating in the plasma of MDS patients was performed using nCounter, a high-throughput platform from NanoString Technologies, resulting in a panel of the following miRNA: miRNA-144, −16, −25, −451, −651, −655 and let-7a, which allows more accurate predictions to be made on the general survival of patients with normal-karyotype MDS [105]. Although some works in this line have been published, additional retrospective and prospective studies are required to see if miRNA expression profiles correspond with prognoses and responses to treatment in patients with MDS.

## 7. Conclusions

Molecular markers gradually become more and more popular in describing MDS; additionally, they discriminate MDS from other BM conditions. The fact is that a universal prognostic scoring system covering all significant MDS parameters has yet to be developed. The main obstacle in the way towards inferring the most accurate prognoses in the development of MDS and suggesting treatment options is insufficient knowledge of the biological aspects of this disease. Therefore, an important task set before clinical oncology is the search for additional molecular-genetic markers with a possibility of integrating them into the existing international prognostic systems, and some of the most promising markers in this respect are miRNA. Current data of the roles of miRNA in MDS suggest that these molecules have the potential to be used as tools for the diagnosis and prognosis of MDS and may have relevance to the response to treatment. Noteworthy, miRNAs are quite promising markers not only because of the biological roles that they have, but also because they will be equally suitable for many different protocols to come. Unlike mRNAs, miRNAs are highly stable, allowing them to be used in retrospective studies for analysis of BM material embedded in paraffin or fixed on coverslips. In conclusions, prognostication will probably evolve in coming years with the large development of targets therapies. Thus, the search for additional prognostic markers for the diagnosis of both de novo MDS and secondary lesions to BM will allow treatment to be personalized in the most precise manner.

## Figures and Tables

**Table 1 medicina-56-00376-t001:** Frequency (~%) of mutations and chromosome abnormalities in myeloproliferative disorders.

Gene	Region	CHIP	MDS De Novo	Secondary AML	t-MDS
*NRAS*	1p13		10	12	
*SF3B1*	2q33	18–35	20–35	8	
*DNMT3A*	2p23	12–18	10–15	10	
*IDH1*	2q33		<5	13	5
*GATA2*	3q21	<5	<5		
*TET2*	4q24	20–25	20–35	17	
	del(5q)		25	6	25
	der (5;17)(p10;q10)		<1		
*EZH2*	7q36	7	5-10	10	
	−7/del(7q)	1–8	10	5–15	35
	der (1;7)(q10;p10)		<1		3
	+8	1–13	20	10	10
*JAK2*	9p24		5	13	
	del(11q)		<5		4
	+11		<1		
*CBL*	11q23	<5	<5		
*ETV6*	12p13	<5	<5		
*KRAS*	12p12		1–10		
	del(12p)/der(12p)		<5		6
	−13/del(13q)		<5		4
	+14/14q		<1		
*IDH2*	15q26		5	9–25	
*SRSF2*	17q25	10	10–20		
*TP53*	17p13	10	10	5	35
*PRPF8*	17p13	1–4	<5		
	iso(17q)/der(17p)		<5		7
	−18				9
*CEBPA*	19q13	1–4	<5		
*ASXL1*	20q11	10–25	10–25	35–40	
	del(20q)		<10	2	5
*RUNX1*	21q22	15	10–15	20	
*U2AF1*	21q22	8–10	<10		
	−21/+21		<1	7	5
*STAG2*	Xq25		<10		
*ZRSR2*	Xp22.2		<10		
	−X		<1		
*ATRX*	X		<1		
	−Y	1–6	<5		2

Clonal hematopoiesis of indeterminate potential (CHIP), myelodysplastic syndromes de novo (MDS de novo), acute myeloid leukemia secondary to MDS (Secondary AML), therapy-related myelodysplastic syndromes (t-MDS).

**Table 2 medicina-56-00376-t002:** Classification systems for MDS.

French-American-British classification (FAB) (1982)	World Health Organization Classification (WHO) (1999–2001–2008–2016)
1. Refractory anemia (RA)	1. MDS with single-lineage dysplasia: refractory anemia (MDS-SLD)
2. Refractory anemia with ringed sideroblasts (RARS)	2. MDS with multilineage dysplasia (MDS-MLD)
3. Refractory anemia with excess of blasts (RAEB)	3. MDS with ringed sideroblasts and single-lineage dysplasia (MDS-RS-SLD)
4. Refractory anemia with excess of blasts in transformation (RAEBt)	4. MDS with ringed sideroblasts and multilineage dysplasia (MDS-RS-MLD)
5. Chronic myelomonocytic leukemia (CMML)	5. MDS with an isolated deletion of the long arm of chromosome 5 (del(5q))
	6. MDS with excess of blasts–1 (MDS-EB-1)
	7. MDS with excess of blasts–2 (MDS-EB-2)
	8. Unclassifiable MDS (MDS-U)
	9. Refractory cytopenia of childhood (RCC)

**Table 3 medicina-56-00376-t003:** Delineation of cytogenetic abnormalities according to their prognostic implications in the IPSS-R.

Very Good	Good	Intermediate	Poor	Very Poor
−Y; del(11q)	normal karyotype; del(5q); double aberrations including del(5q), del(12p), del(20q)	del(7q); +8; +19; i(17q); any other, independent clones	−7; inv(3)/t(3q)/del(3q); double aberration with −7/7q-; complex karyotypes with 3 abnormalities	Complex karyotypes with >3 abnormalities

IPSS-R: Revised International Prognostic Scoring System.

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
