# Peer review of "Prognostic Markers of Myelodysplastic Syndromes"

_medicina, 2020, doi:10.3390/medicina56080376_

Round 1

Reviewer 1 Report

Y.A. Veryaskina and colleagues made a review about cytogenetic and molecular markers with prognostic significance in myelodysplastic syndrome. Overall, the paper is well organized and understandable.

I have no major concern about the manuscript but few suggestions:

-lines 86-88: please rephrase - I think the words "associated with MDS" could be removed to make the sentence clearer for readers.

-table 1: I recommend organizing the table per gene classes instead of chromosomal location as presented in the main text (ie RNA-splicing, DNA methylation, histone modifiers, cohesin, transcription, signaling, tumor suppressors and chromosomal aberrations). It would be easier for readers to distinguish between highly mutated genes (RNA-splicing+++) and unusual mutations (signaling) for example.

-line 148: the sentence suggests that clonal hierarchy in MDS is random. However, it is well known that the order of mutation acquisition is usually biased with RNA-splincing and TP53 mutations being initiating events in most cases while NRAS or KRAS mutations occur in the last events of the disease (Nagata et al, Nature Communications, 2019).

-line 215: the authors emphasize RUNX1 and GATA2 mutations as predisposing factors for MDS. I think they should also cite DDX41 mutations, which are probably more frequent and could define a distinct prognostic subgroup (Lewinsohn et al, Blood, 2016; Sébert el al, Blood, 2019).

-line 336: the term "mutation" should be replaced by "rearrangement" in this sentence.

-Overall, I think that conclusions from case reports should be avoided. Especially, sentences about references 56, 57 and 72 should be removed since conclusions require broader studies to validate the hypotheses raised. 

-In their conclusions, the authors could state that prognostication will probably evolve in coming years with the large development of therapies against targets they discussed (splice inhibitors, luspatercept, APR-246, HDAC inhibitors...). So new studies will be necessary.

Author Response

1) lines 86-88: please rephrase - I think the words "associated with MDS" could be removed to make the sentence clearer for readers.

I deleted this phrase.

2) table 1: I recommend organizing the table per gene classes instead of chromosomal location as presented in the main text (ie RNA-splicing, DNA methylation, histone modifiers, cohesin, transcription, signaling, tumor suppressors and chromosomal aberrations). It would be easier for readers to distinguish between highly mutated genes (RNA-splicing+++) and unusual mutations (signaling) for example.

This table includes not only gene mutations, but also chromosomal abnormalities. Therefore, I decided to create a structure for localization.

3) line 148: the sentence suggests that clonal hierarchy in MDS is random. However, it is well known that the order of mutation acquisition is usually biased with RNA-splincing and TP53 mutations being initiating events in most cases while NRAS or KRAS mutations occur in the last events of the disease (Nagata et al, Nature Communications, 2019).

I added the paragraph:

However, mutations of SF3B1, U2AF1, and TP53 are more likely to be dominant, those of ASXL1, CBL, and KRAS are secondary [Nagata,et al].

Nagata, Y., Makishima, H., Kerr, C.M. et al. Invariant patterns of clonal succession determine specific clinical features of myelodysplastic syndromes. Nat Commun 10, 5386 (2019).

4) line 215: the authors emphasize RUNX1 and GATA2 mutations as predisposing factors for MDS. I think they should also cite DDX41 mutations, which are probably more frequent and could define a distinct prognostic subgroup (Lewinsohn et al, Blood, 2016; Sébert el al, Blood, 2019).

  • I removed the sentence from the text :”It should be noted that although germline mutations in RUNX and GATA2 may predispose to MDS, somatic mutations to the same genes may cause disease progression when myeloid neoplasms are present [43].”
  • The DDX41 is mentioned in the second sentence of this paragraph.

5) line 336: the term "mutation" should be replaced by "rearrangement" in this sentence.

I replaced it.

6) Overall, I think that conclusions from case reports should be avoided. Especially, sentences about references 56, 57 and 72 should be removed since conclusions require broader studies to validate the hypotheses raised. 

[56]  This review presents the frequency of mutations and chromosomal rearrangements. The mutation rates range from less than 1% to over 30%. In this case have been published the data from a rare clinical case. In such articles, the author always says that for statistically reliable results it is necessary to increase the number of cases.

[57], [72]  The main topic of my work is the analysis of synchronous hematologic diseases. Therefore, the link to such works is very interesting and important for me.

Rouslan Kotchetkov, Erin Ellison, Jesse McLean, Bryn Pressnail & Derek Nay (2018) Synchronous dual hematological malignancies: new or underreported entity?, Hematology, 23:9, 596-599, DOI: 10.1080/10245332.2018.1466428

7) In conclusions, prognostication will probably evolve in coming years with the large development of targets therapies.

Reviewer 2 Report

I congratulate the authors on writing this informative review article on the prognostic markers of myelodysplastic syndrome. The article provides a good overview of the landscape of somatic mutations in MDS and its prognostic implications.

Major Comment

The article starts well; however, after initiating the discussion on prognostic systems, the article seems to lose focus and lacks details. It also falls short on providing future directions, and the section on miRNA is rather short and uninteresting. 

Minor comments.

  • Line 45: Do you mean blasts? Blastocyte is an incorrect term.
  • Line 47: “For example, another popular option other than allogeneic hematopoietic stem cell transplantation is epigenetic therapy, with DNA methyltransferase inhibitors as the main drugs; however, the response to treatment is adversely affected by hypomethylating agent resistance” -> Suggest splitting the sentence.
  • Line 123: “C” in the classification does not need to be capitalized.
  • Recommend including the performance of the WHO classification system versus the FAB system in terms of clinical outcomes, response to therapy, and inter-observer agreement. Cite “Howe et al. Blood 2004”.
  • Line 193: Rephrase sentence on EZH2 to a formal statement.
  • Line 222. Secondary MDS occurs with cytotoxic chemotherapy too. For example, Anthracyclines.
  • “ The t-MDS incidence is growing from year to year due to the growing prevalence of tumors and an increase in the number of associated treatment” – Citation needed to support this statement.
  • Cytogenetic abnormalities associated with t-MDS/AML needs further elaboration. On the lines of " There are two types of t-MDS/AML recognized by the World Health Organization classification depending on the causative therapeutic exposure: an alkylating agent/radiation-related type and a topoisomerase II inhibitor-related type. Both differ in their cytogenetic abnormalities. The first type is associated with 5q and 7q, while the latter is associated with balanced translocations involving chromosome bands 11q23 or 21q22."
  • Line 288: Suggest using “highest gene density” versus gene-rich.
  • Since this review includes a discussion on somatic gene mutations in MDS throughout the manuscript, I would suggest having a table with newer prognostic systems that have incorporated such mutations in their scoring systems. Example Nazha et al. Leukemia 2016, Hou et al. Blood cancer journal 2018.

Author Response

1)Line 45: Do you mean blasts? Blastocyte is an incorrect term.

Blastocyte changed to blast cel.

 2) Line 47: “For example, another popular option other than allogeneic hematopoietic stem cell transplantation is epigenetic therapy, with DNA methyltransferase inhibitors as the main drugs; however, the response to treatment is adversely affected by hypomethylating agent resistance” -> Suggest splitting the sentence.

For example, another popular option other than allogeneic hematopoietic stem cell transplantation is epigenetic therapy, with DNA methyltransferase inhibitors as the main drugs; however, the response to treatment is adversely affected by hypomethylating agent resistance. changed to

However, choosing the optimal moment and integrating allo-HSCT into the therapeutic algorithm remains a challenge in many cases. Also the response to treatment is adversely affected by hypomethylating agent resistance

3)Line 123: “C” in the classification does not need to be capitalized.

C changed to c  ^ Line 84,102,105,106,132

4) Recommend including the performance of the WHO classification system versus the FAB system in terms of clinical outcomes, response to therapy, and inter-observer agreement. Cite “Howe et al. Blood 2004”.

In independent validations, the WHO revisions were shown to provide more-homogeneous subgroups of patients and greater prognostic power compared with the FAB system, although controversies remain [Bennett, 2005]. Navarro et al., analyzed the prognostic impact of the WHO and FAB morphologic classifications in a series of patients with primary MDS. They show that 17.7% patients with MDS according to FAB criteria no longer presented MDS after reclassification according to the WHO proposals [Navarro et al.,2006]. In the other study, 103 cases of MDS previously classified by FAB were reclassified according to the WHO proposal. There was a significant interobserver agreement and discrepancies among observers nearly always related to the identification and enumeration of dyspoiesis in neutrophils and megakaryocytes. The present study suggests that in patients with less than 5% blasts, an important negative variable for response to treatment as well as for survival is nonerythroid multilineage dysplasia [Howe et al.2004].

Bennett JM. A comparative review of classification systems in myelodysplastic syndromes (MDS). Semin Oncol. 2005;32(4 Suppl 5):S3-S10. doi:10.1053/j.seminoncol.2005.06.021  

Navarro I, Ruiz MA, Cabello A, et al. Classification and scoring systems in myelodysplastic syndromes: a retrospective analysis of 311 patients. Leuk Res. 2006;30(8):971-977. doi:10.1016/j.leukres.2005.11.015

Howe RB, Porwit-MacDonald A, Wanat R, Tehranchi R, Hellström-Lindberg E. The WHO classification of MDS does make a difference. Blood. 2004;103(9):3265-3270. doi:10.1182/blood-2003-06-2124

5) Line 193: Rephrase sentence on EZH2 to a formal statement.

Changed to

 EZH2 is located in 7q36.1, which is normally deleted in MDS, and that is how this gene has lost its function: either due to a mutation or due to the chromosomal deletion.

6) Line 222. Secondary MDS occurs with cytotoxic chemotherapy too. For example, Anthracyclines.

Added to text with reference.

7) “ The t-MDS incidence is growing from year to year due to the growing prevalence of tumors and an increase in the number of associated treatment” – Citation needed to support this statement.

This was my guess. I removed the sentence from the text.

8) Cytogenetic abnormalities associated with t-MDS/AML needs further elaboration.

On the lines of " There are two types of t-MDS/AML recognized by the World Health Organization classification depending on the causative therapeutic exposure: an alkylating agent/radiation-related type and a topoisomerase II inhibitor-related type. Both differ in their cytogenetic abnormalities. The first type is associated with 5q and 7q, while the latter is associated with balanced translocations involving chromosome bands 11q23 or 21q22."

There are two types of t-MDS/AML recognized by the WHO classification depending on the causative therapeutic exposure: an alkylating agent/radiation-related type and a topoisomerase II inhibitor-related type. Both differ in their cytogenetic abnormalities. The first type is associated with involving chromosomes 5 (−5/del(5q)) and 7 (−7/del(7q)), while the latter is associated with balanced translocations involving chromosome bands 11q23 or 21q22.

         Bhatia S. Therapy-related myelodysplasia and acute myeloid leukemia. Semin Oncol. 2013 Dec;40(6):666-75. doi: 10.1053/j.seminoncol.2013.09.013.

9) Line 288: Suggest using “highest gene density” versus gene-rich.

gene-richest changed to highest gene density

10) Since this review includes a discussion on somatic gene mutations in MDS throughout the manuscript, I would suggest having a table with newer prognostic systems that have incorporated such mutations in their scoring systems. Example Nazha et al. Leukemia 2016, Hou et al. Blood cancer journal 2018.

I added the paragraphs:

Gene mutations have not yet been included in the 2016 WHO classification and IPSS-R. Mutational analyses showed that mutations of CBL, IDH2, ASXL1, DNMT3A, and TP53 were independently associated with shorter survival. Patients within each IPSS-R or 2016 WHO classification-defined risk group could be stratified into two risk subgroups based on the mutational status of these five genes; patients with these poor-risk mutations had an OS shorter than others in the same risk group, but similar to those with the next higher risk category [Hou, et al.].

The IPSS-R was developed for untreated MDS patients based on clinical data. Nazha et al., created a new model that incorporates mutational data to improve the predictive capacity of the IPSS-R in treated MDS patients. Independent significant prognostic factors for survival included age, IPSS-R and mutations EZH2, SF3B1, TP53, which associated with worse OS [Nazha, A].

Nazha, A., Narkhede, M., Radivoyevitch, T. et al. Incorporation of molecular data into the Revised International Prognostic Scoring System in treated patients with myelodysplastic syndromes. Leukemia 30, 2214–2220 (2016). https://doi.org/10.1038/leu.2016.138

Hou, H., Tsai, C., Lin, C. et al. Incorporation of mutations in five genes in the revised International Prognostic Scoring System can improve risk stratification in the patients with myelodysplastic syndrome. Blood Cancer Journal 8, 39 (2018). https://doi.org/10.1038/s41408-018-0074-7

The section on microRNA is really small because it requires a separate review. I just wanted to point out that there are both genetic and epigenetic prognostic markers.